# Transformers to Predict the Applicability of Symbolic Integration Routines

**Rashid Barket**
Coventry University
`barketr@coventry.ac.uk`

**Uzma Shafiq**
Coventry University
`shafiqu9@coventry.ac.uk`

**Matthew England**
Coventry University
`Matthew.England@coventry.ac.uk`

**Jürgen Gerhard**
Maplesoft
`jgerhard@maplesoft.com`

## Abstract

Symbolic integration is a fundamental problem in mathematics: we consider how machine learning may be used to optimise this task in a Computer Algebra System (CAS). We train transformers that predict whether a particular integration method will be successful, and compare against the existing human-made heuristics (called guards) that perform this task in a leading CAS. We find the transformer can outperform these guards, gaining up to 30% accuracy and 70% precision. We further show that the inference time of the transformer is inconsequential which shows that it is well-suited to include as a guard in a CAS. Furthermore, we use Layer Integrated Gradients to interpret the decisions that the transformer is making. If guided by a subject-matter expert, the technique can explain some of the predictions based on the input tokens, which can lead to further optimisations.

## 1 Introduction

Symbolic integration is well-studied, proven to be undecidable (see e.g. discussion in Chapter 22 in Gerhard and Von zur Gathen [2013]). The most widely applicable method is the Risch algorithm (Risch [1969]). However, the original algorithm does not work with special functions, and implementations of this algorithm has trouble with algebraic extensions. It is also complicated, taking more than 100 pages to explain in (Geddes et al. [1992]): no Computer Algebra System (CAS) has a full implementation. Hence, researchers in CA have designed a variety of other symbolic implementation methods.

More recently have there been attempts to use Machine Learning (ML) to perform symbolic integration. First, Lample and Charton [2020] implemented a transformer to directly integrate an integrand, outperforming several mature CASs on the generated test data. Further attempts have been made using LLMs (Noorbakhsh et al. [2021]), and by chaining explainable rules (Sharma et al. [2023]). More generally, there has been work developing a single LLM to perform a variety of mathematical reasoning tasks including (in)definite integration such as (Drori et al. [2022], Hendrycks et al. [2021]).

We are interested in improving the efficiency of indefinite integration within a CAS using ML, while still ensuring that correctness of the answer is guaranteed, through ML-based optimisation and algorithm selection. Recent work applied TreeLSTMs to show there is room for significant performance improvements (Barket et al. [2024a]). We focus on the popular commercial CAS, Maple. Maple's user-level integration call is essentially a meta-algorithm: it can employ several different **methods** for symbolic integration. They are currently tried in a deterministic order until one succeeds, at which point the answer is returned without trying the other methods. A key part of this

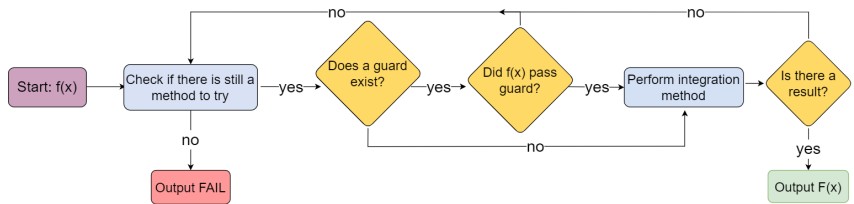

Figure 1: A high-level overview of how the indefinite integral command works in Maple to calculate $F(x) = \int f(x)dx$.

implementation is what we call the **guards**: code that is run prior to calling one of the methods to decide whether or not attempting to use the method is worthwhile. The reasoning behind having a guard is that some of the methods are computationally expensive: it is a waste of time to go through a complex algorithm just to find out that it fails. Figure 1 gives a high-level overview of this idea.

Not all methods have a guard, and some guards do partial work for the method making them expensive for a quick check. Appendix A gives a list of methods and their guard (should they exist). We create small ML models to predict success for methods and investigate whether these can provide guards for methods that lack them, or replace the computationally expensive guards (provided they do better). We also investigate whether interpretations of these models might inform further guard development.

## 2 Dataset

Our dataset is composed of integrable expressions in their prefix notation. The data comes from six different data generators: the FWD, BWD, and IBP generators from Lample and Charton [2020]; the RISCH generation method from Barket et al. [2023]; the SUB generation method from Barket et al. [2024a]; and the LIOUVILLE generation method from Barket et al. [2024b]. For more information see Appendix B.1. We obtain the labels for each integrand by recording which methods succeed (1) and which fail (0) for each expression. We note that this labelling is considerably more computationally expensive than the ML training later. The data is stored as a list where, for position $i$ in the list, the value records if method $i$ is a success or failure.

The data goes through several pre-processing steps to shrink the vocabulary size and to avoid having expressions of similar form over-represented in the dataset, as described in Appendix B.2. In total, there are 1.5M and 60K samples for training and testing respectively, with an equal split from each data generator. The labels do not occur equally however: some methods have a much higher rate of success than others. This is because some of the methods are only made for certain types of integrands (e.g. Elliptic) whereas others are more general-purpose (e.g. Risch). Figure 2 shows the frequency of integrating successfully over the train set from each method.

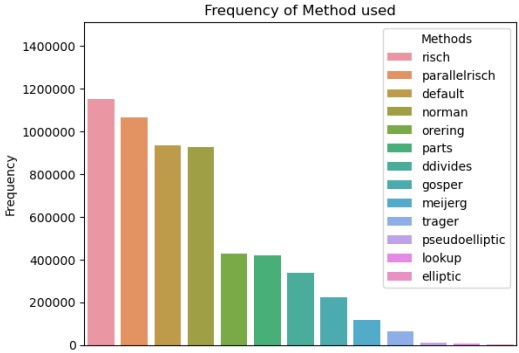

Figure 2: Frequency of each method being successful on 1.5M examples from six data generators.

Table 1: Comparing each transformer to predicting positive every time on the test dataset. These methods have no guard; they always run when Maple's algorithm reaches that specific method.

| Method | Accuracy (%) | |
|---|---|---|
| | Transformer | Guard |
| `Default` | **94.86** | 82.15 |
| `DDivides` | **94.13** | 28.18 |
| `Parts` | **93.10** | 37.05 |
| `Risch` | **94.53** | 89.35 |
| `Norman` | **95.74** | 71.67 |
| `Orering` | **97.21** | 37.88 |
| `ParallelRisch` | **97.82** | 82.73 |

Table 2: Comparing each transformer to the Maple guard on the full test dataset.

| Method | Accuracy (%) | | Precision (%) | |
|---|---|---|---|---|
| | Transformer | Guard | Transformer | Guard |
| `Trager` | **98.21** | 67.55 | **92.21** | 15.88 |
| `MeijerG` | **96.78** | 88.72 | **89.58** | 57.78 |
| `PseudoElliptic` | **99.28** | 62.61 | **62.86** | 2.03 |
| `Gosper` | **94.04** | 92.51 | **92.08** | 80.21 |

# 3 Training Transformers to Predict Method Success

## 3.1 Experiment Setup

For each of the methods in Maple 2024, we train a classifier to determine whether it is worthwhile attempting said method. We use an encoder-only transformer Vaswani et al. [2017] to predict success or failure, with architecture similar to Lample and Charton [2020] and Sharma et al. [2023]. Hyper-parameters and the full model architecture are discussed in Appendix C. The results are then compared to the guards implemented by human experts in Maple. In this scenario, a false positive is considered worse than a false negative. This is because trying a method and failing wastes compute time, whereas skipping an algorithm that may succeed may be okay because another method could possibly succeed (and if none do, we could always then try the guarded ones at the end). Thus, we evaluate accuracy and precision when comparing each transformer to each guard.

## 3.2 Results

Table 1 shows the results for methods that do *not* have a guard. In this case, the transformers are compared to a hypothetical guard that simply predicts a positive label every time, since the current workflow would always attempt such methods. These transformers range in accuracy from $93\%$ to $98\%$. Hence, there is much scope for using ML to prevent wasteful computation. An example of such a case is for `Risch` and `Norman`: both have an operation, `Field`, that creates the base field along with its field extensions before proceeding with the algorithm. This operation takes nearly $20\%$ of their runtime before determining whether to run the rest of the algorithm or output fail. On a batch of 1024 examples, the mean transformer runtime was only $0.0895$s compared to `Field` which took $0.125$s ($39.7\%$ longer). In such a case, the transformers can help save computation time before performing the `Field` operation.

Table 2 shows the results for the methods that do already have a guard. In each of these cases, we see that a transformer can beat the guards, and some by a margin of over $30\%$. We note that the guards for `Trager` and `PseudoElliptic` are particularly weak compared to the transformers, with much lower accuracy and precision. The transformers can avoid the false positives that the guards cannot. The number of positive samples for `PseudoElliptic` is scarce and future work may include an anomaly detection approach for this method. A caveat is that the inference time of the transformers is higher than the existing guards. As mentioned above, the transformers took $8.95\mathrm{e}{-2}$s on average per sample whereas `Trager`, `MeijerG`, `PseudoElliptic`, and `Gosper` took $1.1\mathrm{e}{-6}$, $1.4\mathrm{e}{-6}$, $4.7\mathrm{e}{-7}$, and $4.7\mathrm{e}{-7}$s respectively. The transformers take magnitudes longer to complete inference. However, we hypothesize that the huge gains in precision over these guards will help prevent a lot of wasteful

Table 3: Comparing each transformer to a perfect Maple guard on the filtered dataset.

| Method | # Samples | Accuracy (%) | | Precision (%) | |
|---|---|---|---|---|---|
| | | Transformer | Guard | Transformer | Guard |
| Trager | 19287 | **95.37** | 15.88 | **93.39** | 15.88 |
| PseudoElliptic | 19287 | **98.04** | 2.03 | **63.37** | 2.03 |
| Gosper | 18929 | **86.26** | 80.21 | **96.13** | 80.21 |

computation and so overall, Maple's integration algorithm would save time, to be confirmed in future work.

Some guards are known to be perfect at predicting incorrectness: i.e. if such a guard outputs that a method will not work, then it is a mathematical proof that it will never succeed. Although such guards will always predict the true negatives correctly, they may make mistakes on the true positives. Table 3 reports performance after we filter the data with such guards, by removing the samples the guard finds negative (the first two have a similar guard which caused them to have the same filtered dataset). Note that because these guards are perfect at predicting the negative target class, their accuracy and precision in Table 3 are exactly that of the precision on the full dataset in Table 2. We see that, unlike the guards, the transformers have good performance on both classes.

# 4 Layer Integrated Gradients to Interpret the Classifiers

We have seen that ML can optimise the work of CASs here, but we are also interested in *how* the ML models make their decisions: both to give confidence in their results and perhaps to inform better hard-coded guards. This proved to be the case when SHAP values were used to explain the predictions of traditional ML models optimising Cylindrical Algebraic Decomposition (Pickering et al. [2024]). However, transformers are most widely used in Math-AI Zhou et al. [2024], Sharma et al. [2023], Charton [2024] and their explanation remains a challenge. Sharma et al. [2023] proposed an explainable approach for symbolic integration: computing integrals by applying rules step-by-step to form a chain from input to output. However, while this explains how the integral may be computed, it does not explain the decisions the transformers make to select these rules.

We further experiment with interpreting the transformers using Layered Integrated Gradients (LIGs) Sundararajan et al. [2017]; an extension of Integrated Gradients that gives insight into different layers of a neural network. We used LIGs with the embedding layer of our model to compute the attribution score of each input token as the embedding layer converts the input tokens into a dense vector space. We used 50 steps and the default baseline as described in Kokhlikyan et al. [2020].

One interesting observation concerns the `Risch` method and the attribution scores found for the token of the absolute value function (`abs`). These always contribute negatively, usually highly so, in the observed samples which indicates that the presence of this token will adversely impact the prediction for use of this method. A visualisation for a particular example is shown in Figure 3. After discussing with domain experts, we find this a satisfying interpretation: the Risch algorithm is suited for elementary functions and the absolute value function does not fall in this category. In fact, the Risch algorithm is not proven to work when the absolute value function is included in the field defined in the algorithm (Gerhard and Von zur Gathen [2013]). Note this explanation also suggests a simple edit to improve the existing guard code!

The attribution scores for all the samples were calculated and aggregated to plot the graph in Figure 4. It can be observed that the attribution scores for `abs` are the most negative, further demonstrating the alignment of the interpretation and the human expert. Of course, this is just one observation for a single method and domain expertise remains crucial for validating the insight. But it points to potential for further progress through XAI tools.

[CLS] mul INT+ CONST1 mul x pow abs cos mul INT+ 2 pow x INT+ 2 INT- 1

Figure 3: Example sequence to depict the attribution scores corresponding to different tokens where blue is positive and red is negative. Note the strongly negative score for the `abs` token.

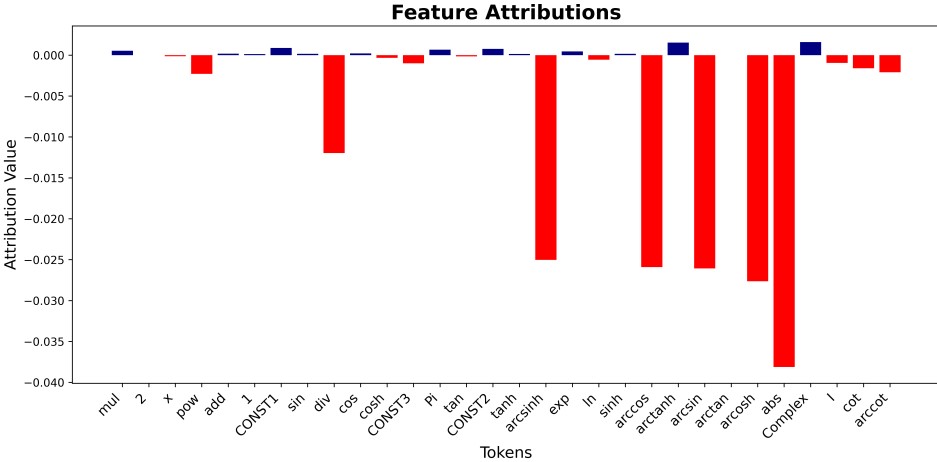

Figure 4: Aggregated attribution scores for all test samples containing the `abs` token for the Risch method. Note the `abs` token has a strongly negative score.

## 5 Conclusions

In summary, we have demonstrated how we may improve the symbolic integration function in Maple without risking its mathematical correctness, by using transformers to predict when an integration method in Maple will succeed. The transformers were compared to heuristics crafted by domain experts: Table 1 shows that when a heuristic does not exist, a transformer is a suitable guard at predicting success, achieving between 93-98% accuracy. For the four methods that do have a guard, we also show that transformers make better predictions than these guards. In some cases, accuracy and precision increase significantly compared to the guards, over 30% and 70% respectively.

The models, specifically the embedding layers, were then analysed using LIGs to try to interpret the models. We demonstrated that this approach can give a satisfying explanation for some predictions (the presence of the `abs` token being heavily associated with a negative label for the `Risch` method) although we note that domain knowledge is needed to understand why LIGs produce this result. This explanation in turn suggests improvements to the human-designed guards.

### 5.1 Future Work

We have shown the potential for ML optimisation of the Maple integration meta-algorithm. As next steps we will embed the transformers and evaluate the savings: measured both in number of methods tried (by trying them in order of probability of success) and in overall CPU time.

The existence of some perfect guards suggests there is scope for a hybrid approach between algebraic tools and transformers which we will also explore in future work. After that, we will seek to implement our best approach into the actual workflow of Maple's symbolic integration algorithm. We would then be able to empirically measure how much time is saved in the algorithm rather than analysing each method individually.

We will also continue to experiment with XAI tools to see if the transformers are learning the same rules as the integration method it is learning about, inspired by the insight observed above. The LIGs only offered explanations at a token level and most likely we need a tool that can also analyse the presence and prevalence of sequence and groups of tokens. While LIGs provided us with some insight, there was a need for a domain expert to validate this, and that is likely to remain the case.

Finally, we note again that in similar applications such as Pickering et al. [2024] and Coates et al. [2023], the authors were able to discover new interpretable heuristics based on the observations from the ML models. Being able to understand the predictions is interesting, but the ultimate goal would be to discover new interpretable heuristics for the methods that do not have a guard using XAI tools. This can give better insight on how we understand symbolic integration as a whole and discover new mathematical properties associated with these methods.

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

# A Integration Methods and their Guards

There are 13 different integration methods exposed to the user in Maple 2024. These can be passed to the `method` parameter in the `int` function. We obtain results for all sub-routines simultaneously by passing `method=_RETURNVERBOSE` to the command. An example screenshot is shown in Figure 5. Running this process over the entire dataset is how we generate the labels for the experiment.

```
> int(sin(x), x, method=_RETURNVERBOSE)
```
$$\left[ \text{"lookup"} = -\cos(x), \text{"default"} = -\cos(x), \text{"norman"} = -\frac{2}{1+\tan\left(\frac{x}{2}\right)^2}, \text{"meijerg"} = \sqrt{\pi}\left(\frac{1}{\sqrt{\pi}} - \frac{\cos(x)}{\sqrt{\pi}}\right), \text{"risch"} = -\cos(x), \right.$$
$$\left. \text{"parallelrisch"} = -\cos(x) - 1, \text{"orering"} = -\cos(x), FAILS = (\text{"gosper"}, \text{"derivativedivides"}, \text{"trager"}, \text{"elliptic"}, \text{"pseudoelliptic"}, \text{"parts"}) \right]$$

Figure 5: An example output from the indefinite integration command when all the methods are called. Some answers have a different form but are all mathematically equivalent.

Exposing the guards is a more subtle issue. Within Maple there exists the package `IntegrationTools:-Indefinite` which lets the user gain access to the indefinite integration commands, including each method listed in Figure 5 individually. With the Maple command `showstat()`, the user can then see (some of the) source code for them. The list of guards can be found in the GitHub link (provided after reviews) but we give a plain-language explanation of each one here:

- `Trager`: check if the expression is a radical algebraic function.
- `MeijerG`: Check if the expression is a special function or algebraic. Our dataset includes no special functions so we only use the latter condition.
- `PseudoElliptic`: Check if the expression is a radical function.
- `Gosper`: Using the `DETools` package, check if the expression is hyperexponential. A function $H$ is hyperexponential of $x$ if

$$\frac{\frac{d}{dx}H(x)}{H(x)} = R(x), \text{a rational function of } x$$

For more information on the meaning of the data types described, see Maple's type help page: `https://maplesoft.com/support/help/maple/view.aspx?path=type`.

# B Data Generation and Preprocessing

## B.1 Generation Methods

Lample and Charton [2020] were the first to describe data generation methods for indefinite integration. They presented three methods to produce (integrand, integral) pairs:

- **FWD:** Integrate an expression $f$ through a CAS to get $F$ and add pair $(f, F)$ to the dataset.
- **BWD:** Differentiate $f$ to get $f'$ and add the pair $(f', f)$ to the dataset.
- **IBP:** Given two expressions $f$ and $g$, calculate $f'$ and $g'$. If $\int f'g$ is known (i.e. exists in a database) then the following holds (integration-by-parts): $\int fg' = fg - \int f'g$. Thus, we add the pair $(fg', fg - \int f'g)$ to the dataset.

While this can generate a sufficient quantity of data, there exists a problem with insufficient variety of data. The present authors and others have discussed the issues (Davis [2019], Piotrowski et al. [2019], Barket et al. [2023]). We briefly overview the problems here:

- There is a bias in the length of the expressions where a generator can produce long integrands and short integrals or vice-versa.
- Generated integrands can end up having too similar of a form within the same dataset (i.e. differing only by their coefficients).

- They have a hard time generating integrands of certain forms (see Barket et al. [2024b] for more on this point).

Prior work by the authors creates several different generators to help address such issues based on mathematical properties of integrands. Liouville is the first to have described the general form of an elementary integral if it exists. Two of the generators are based on the work of Liouville so we first state Liouville's theorem (Section 12.4 in Geddes et al. [1992]) for symbolic integration.

**Theorem 1 (Liouville's theorem)** *: Let $D$ be a differential field with constant field $K$ (whose algebraic closure is $L$). Suppose $f \in D$ and there exists $g$, elementary over $D$, such that $g' = f$. Then, $\exists\, v_0, \ldots, v_m \in D, c_1, \ldots, c_m \in L$ such that*

$$f = v_0' + \sum_{i=1}^{m} c_i \frac{v_i'}{v_i} \implies g = v_0 + \sum_{i=1}^{m} c_i \log(v_i).$$

The two generators based on Liouville's theorem are:

- **RISCH:** (Barket et al. [2023]) Reverse engineer the steps of the Risch algorithm, whose correctness is underpinned by Liouville's Theorem, which outputs the integral of a given input. We generate an expression $f = \frac{R}{b}$ such that $b$ is a fixed expression and $R$ are represented as symbolic coefficients. We then follow the steps of the Risch algorithm to determine what values these symbolic coefficients can be in order to satisfy integrability. This guarantees $f$ is integrable and the Risch algorithm outputs the integral of $f$.

- **LIOUVILLE:** (Barket et al. [2024b]) Similar to the BWD method but follows the form of Liouville's theorem. Generate a random rational expression $\frac{N}{D}$ and also generate two sums of logarithms $A = \sum_{i=0}^{j} c_i \log(a_i)$ and $A = \sum_{i=0}^{k} d_i \log(b_i)$. The key difference between $A$ and $B$ are that $a_i$ are from the factors of $D$ and $b_i$ are *not* factors of $D$. Then we set $f = \frac{N}{D} + A + B$ and we add $(f', f)$ to our dataset.

Finally, similar to IBP, we also have one more data generator based on the substitution rule from calculus and an existing dataset.

- **SUB**: (Barket et al. [2024a]) Given $f$ and $g$, calculate $g'$ and let $u = g(x)$. If $\int f$ is already known, then the following holds (substitution rule): $\int f(g(x))g'(x)dx = \int f(u)du$. We add $(f(u), \int f(u))$ to our dataset.

## B.2   Data Preprocessing

The data preprocessing steps are primarily the same as described in Barket et al. [2024a] and we list them here for ease of access.

Each number is preceded by an `INT+` or `INT-` token to denote whether the integer being encoded is positive or negative. Then, we replace the actual numbers in the following manner:

1. if the number is $0, 1$ or $2$, then create a corresponding token;

2. if the number is a single-digit number that is not $0, 1$ or $2$, replace the number with a `CONST` token;

3. if the number has two digits, replace by a `CONST2` token; and

4. for all other cases, replace by a `CONST3` token.

As this is a classification task, the value of the constant does not (usually) matter and this helps simplify the data without losing much information. We keep the integers in the range $[-2, 2]$ as they can end up being important (e.g. $(x+1)^1$ and $(x+1)^{-1}$ integrate very differently). If any integrands match after this transformation, we only keep one unique copy to ensure a variety of expressions in the data. Lastly, we follow Devlin et al. [2019] and include a `[CLS]` token at the start of every sequence as a representation of the entire sequence. The output of this token after going through the encoding layers is used for the classification task.

# C  Model Architecture and Code

All data used in the paper and all code to generate the figures and results are available on Zenodo: zenodo.org/records/14013787. We also provide the Github link for the wider project: github.com/rbarket/TransformersVsGuards.

The Zenodo link is all the current code used for this paper which will remain fixed. At the the GitHub link we will continue to grow the dataset size, make small updates to the algorithm, and continue to add more elementary and special functions to the generator.

One of the goals in Section 3 was to make a smaller transformer than seen in previous literature for similar tasks (Sharma et al. [2023], Lample and Charton [2020]). This is because the task is conceptually easier (binary classification), and this would also lead to faster inference times. To this end, the model was an encoder-only transformer with 4 heads and 4 layers with dimension 128. We include a dropout layer with $p = 0.3$, and a linear layer with sigmoid activation at the final step. We used the Adam optimizer, BCE loss, and the LR scheduler `ReduceLRonPlateau(factor=0.05, patience=3)`. The batch size was 256. Data ranged from 1 to 1012 tokens in prefix notation. The model is trained on two Quadro RTX 8000 GPU's. Each model is trained for 100 epochs with early stopping implemented. We also checked the inference time of the model and a batch of 256 samples took on average $0.0495$s over 10 runs using Pytorch's Benchmark Timer functionality.

The Layer Integrated Gradients in Section 4 had two parameters. First was the number of splits which was kept 50 and second was the baseline vector which was left to the default value.

