# OpenReview forum: "Transformers to Predict the Applicability of Symbolic Integration Routines"
_NeurIPS.cc/2024/Workshop/MATH-AI — MATH-AI 24_

### Official Review · Reviewer_T4Cp · 2024-10-03
**Review for Submission75**

**Rating:** 3
**Confidence:** 4

**Review:**

Summary: Authors show that it's possible to train a ML model to predict which integration technique will be more powerful when a software tries to find an antiderivative for a function $f$.

Strength:
- This method can help software to decide which integration technique to use.
- Results compared to Maple sound pretty strong.

Weakness:
- To me, it looks like the bulk of the work is on the data collection which comes from prior work and it's not novel at all.
- There is very little understanding on how the model make any decision. It's clear that abs plays a role, but there is no additional explanation after this and this would be interesting from the interpretability point of view.
- There is little explanation apart from results.

---

### Official Review · Reviewer_dqTW · 2024-10-07
**This paper explores the use of a transformer model to optimize symbolic integration within Computer Algebra Systems (CAS), and outperforms the established heuristics (guards) in Maple, suggesting a potentially more efficient method for symbolic computations.**

**Rating:** 7
**Confidence:** 4

**Review:**

This paper explores the use of a transformer model to optimize symbolic integration within Computer Algebra Systems (CAS), particularly by training the model to predict the success of various integration methods, comparing its performance against traditional human-made heuristics ("guards") in Maple. The transformer model outperforms these established heuristics, suggesting a shift towards machine learning-driven strategies in automating symbolic computations.

Pros:
1. A very valuable effort to implement transformer network on predicting whether a integration method is worthy to use.

2. A large dataset of integrable expressions is provided, which will benefit for ML on symbolic integration.

Cons:
1. More novel contributions should be emphasized in this work.

2. As the author said in the section of future work, more extensive and complete experiments should be performed to further investigate the advantages.

---

### Official Review · Reviewer_p4zs · 2024-10-08
**The paper is overall well-written and the main ideas are clearly explained.**

**Rating:** 6
**Confidence:** 2

**Review:**

Paper Summary:
This paper considers how machine learning may be used to optimize this task in a Computer Algebra System. The authods train a transformer that predicts whether particular integration methods will be successful, compare the existing human-made heuristics, and find the transformer can outperform these methods.

Weakness and Strength:

(+) The paper is well-structured and clearly presents the problem, methodology, and findings.

(+) This paper studies an interesting and well-motivated research question: how machine learning may be used to optimise this task in a Computer Algebra System (CAS).

(-) It would be beneficial to briefly highlight the implications of your findings. For example, explain how the ability of the transformer to outperform existing guards impacts the field of symbolic integration or CAS.

---

### Official Review · Reviewer_Vvfp · 2024-10-09

**Rating:** 7
**Confidence:** 4

**Review:**

Strengths:
* Demonstrates an interesting application of transformers, i.e., to predicting the applicability of symbolic integration routines
* Successfully demonstrates it outperforms human-engineered heuristics

Weaknesses:
* Evaluation can benefit from more complete reporting (see suggestions below)
* Lack of evaluation in the context of the extra cost incurred by running the Transformer guard vs. the original guard. Lack of evaluation of total savings of solving actual integration problems due to the Transformer guard (although it’s said that this is future work).

Questions:
* Line 9: “Symbolic integration is well-studied, proven to be #P-complete (Kawamura [2011]).” The reference seems to be about numerical integration and makes no mention about symbolic integration? Also, it is unclear to me how symbolic integration can be formulated as a counting problem, a prerequisite for talking about whether a problem is #P-complete.
* Why is atan missing in Figure 4?

Suggestions:
* Line 23: “Barket et al. [2024a]” -> “(Barket et al. [2024a])”
* In Table 1, for the methods that don’t have guards, I recommend comparing with the “trivial” baseline guard that just always says yes (which is effectively what happens for methods without guards in Maple). Otherwise it is difficult to argue that having the transformer guard in these cases would be helpful at all vs. simply not having a guard.
* Although you argued precision is more important than recall, I think it would still be helpful to report recall to give the reader a more comprehensive picture. For example, you say in the text that some guards have perfect recall—it would be helpful to see that in the table.
Writing in Section 4 can be improved. Some citations in Section 4 lack parentheses and there are typos and grammatical mistakes.
* It may be helpful to explain a bit of background on LIGs in Section 4 (or at least refer to an appendix that provides more background on it)

---

### Decision · Program_Chairs · 2024-10-09

Accept